# Diagnostic Considerations of Intermetatarsal Bursitis: A Systematic Review

**DOI:** 10.3390/diagnostics13020211

**Published:** 2023-01-06

**Authors:** Sif Binder Larsen, Stinne Byrholdt Søgaard, Michael Bachmann Nielsen, Søren Tobias Torp-Pedersen

**Affiliations:** 1Department of Diagnostic Radiology, Rigshospitalet, 2100 Copenhagen, Denmark; 2Department of Biomedical Sciences, University of Copenhagen, 2200 Copenhagen, Denmark

**Keywords:** intermetatarsal bursitis, forefoot pain: metatarsalgia, Morton’s neuroma, forefoot bursa, bursitis

## Abstract

Intermetatarsal bursitis (IMB) is an inflammation of the intermetatarsal bursas. The condition causes forefoot pain with symptoms similar to those of Morton’s neuroma (MN). Some studies suggest that IMB is a contributing factor to the development of MN, while others describe the condition as a differential diagnosis. Among patients with rheumatic diseases, IMB is frequent, but the scope is yet to be understood. The aim of this paper was to investigate the diagnostic considerations of IMB and its role in metatarsalgia by a systematic review approach. We identified studies about IMB by searching the electronic databases Pubmed, Embase, Cochrane Library, and Web of Science in September 2022. Of 1362 titles, 28 met the inclusion criteria. They were subdivided according to topic: anatomical studies (*n* = 3), studies of patients with metatarsalgia (*n* = 10), and studies of patients with rheumatic diseases (*n* = 15). We conclude that IMB should be considered a cause of pain in patients with metatarsalgia and patients with rheumatic diseases. For patients presenting with spreading toes/V-sign, IMB should be a diagnostic consideration. Future diagnostic studies about MN should take care to apply a protocol that is able to differ IMB from MN, to achieve a better understanding of their respective role in forefoot pain.

## 1. Introduction

Intermetatarsal bursitis (IMB) is an inflammation of the intermetatarsal bursas located between the metatarsal heads, dorsal to the deep transverse metatarsal ligament [1]. The symptoms of IMB are very similar to those of Morton’s neuroma, with forefoot pain characterized by a sharp, stinging pain that worsens by walking and is often accompanied by numbness in the adjacent toes. The typical location of IMB is the 2nd and 3rd interspace [2].

IMB and Morton’s neuroma are described both as differential diagnoses, and IMB as a contributing factor to the development of Morton’s neuroma [3,4]. The intermetatarsal bursa is suggested to cause Morton’s neuroma in two ways; one suggests the intermetatarsal bursa as a part of the compression theory: swelling and compression of the bursa causes it to bulge and compress the neurovascular bundle [5], the other suggests that inflammation in the intermetatarsal bursa causes the fibrotic changes in the nerve [1]. Within the last decade, IMB has drawn more attention in the rheumatologic field, where an association between IMB and systemic inflammatory diseases such as rheumatoid arthritis and systemic lupus erythematosus is being investigated [6,7].

In the literature on forefoot pain and Morton’s neuroma, IMB occurs repeatedly. However, little consistency is found on the implications of IMB and additionally, inadequate literature exists on how to distinguish the two conditions. The orthopaedic surgeon or radiologist who deals with these diseases daily might know how to differ the two conditions, but limited studies on the subject make it hard to reproduce and standardize these observations. The role of IMB in metatarsalgia is not evident.

The aim of this paper is to investigate the diagnostic considerations of IMB and its role in metatarsalgia by conducting a systematic review to identify anatomical, histopathological, diagnostic, and treatment studies that include the intermetatarsal bursa or IMB. Furthermore, to discuss the potential implication for future diagnostic and treatment strategies.

## 2. Materials and Methods

This study was carried out according to the Preferred Reporting Items for Systematic Reviews and Meta-Analysis (PRISMA) [8]. The review was not registred, but the protocol is available upon request to corresponding author. Our study was designed to answer the following ‘PICO’: Participants were patients with metatarsalgia or asymptomatic; Interventions included any intervention; Comparisons were not specified; and Outcome included results about IMB or the intermetatarsal bursa.

Search strategy: We conducted a literature search in the following databases: Pubmed, Embase, Cochrane Library, and Web of Science. The search was conducted by Sif Binder Larsen with the assistance of a hospital librarian, and last updated in September 2022. No limits or filters were applied in the searches, e.g., publication year or human studies.

Descriptions of IMB were primarily found in literature with a broader theme or different topic, e.g., Morton’s neuroma. Based on this, we performed a wide search to include all studies on IMB and Morton’s neuroma. The search strategy included three main designations of these pathologies with any possible variations of these: Morton neuroma, intermetatarsal neuroma, and IMB. Detailed search strategies are found in Appendix A.

In- and exclusion criteria: The following criteria were applied in the selection of studies: Inclusion: studies that mention intermetatarsal bursa/bursitis in the title and/or abstract, including studies that use any phrase that refers to the same anatomical structure or studies with a focus on differential diagnoses to Morton’s neuroma. Exclusion: studies including children (<18-years-old), veterinary studies, non-original studies, studies without full text accessible, or studies not available in an English translation.

Study selection and data extraction: Title and abstract screening and screening of full text for eligibility were done by two independent reviewers (Sif Binder Larsen and Stinne Byrholdt Søgaard) according to the predefined in- and exclusion criteria. Cases of disagreement were resolved by consensus in each step. In three cases of disagreement during the full-text screening, a third, senior assessor made the decision.

A total of 1362 articles were identified from the databases after duplicates were removed. A total of 88 proceeded to full-text review; of these 28 were included in the review. An outline of the review process and exclusion reasons are shown in Figure 1.

The following variables were extracted from the papers: First author, publishing year, type of study, design, the aim of study, study population, number of participants, female:male ratio, and mean age with (range) or ±standard deviation. The included articles were subdivided into three groups: anatomical studies (*n* = 3), studies on patients with metatarsalgia (*n* = 11), and studies on patients with autoimmune disorders, e.g., rheumatoid arthritis (*n* = 15). One study was placed in both the anatomical and the metatarsalgia group, because it was multipurpose. Results regarding IMB for each study are found in Appendix A.

Risk of bias: We assessed the risk of bias of the included studies using the Quality Assessment of Studies of Diagnostic Accuracy Studies (QUADAS-2) tool [9]. Due to the heterogeneity of the included studies, we chose the assessment tool that fits the majority of the studies. QUADAS-2 consists of four key domains: patient selection, index test, reference standard, and flow and timing. Each domain is assessed for risk of bias and the first three regard applicability. The risk of bias was carried out with assistance from the QUADAS-2 background document available from https://www.bristol.ac.uk/population-health-sciences/projects/quadas/quadas-2/, (accessed on 29 September 2022).

## 3. Results

### 3.1. Study Characteristics

#### 3.1.1. Anatomical Studies

Three studies investigated the anatomic relations of the intermetatarsal bursas in cadaveric feet. An overview of the results is available in Table 1 (a). The presence of bursas in the second and third interspace and their distal extent to become interphalangeal was consistent in all three studies. The proximity to the neurovascular bundle in these interspaces is emphasized.

#### 3.1.2. Patients with Metatarsalgia

Eleven studies investigated patients with metatarsalgia and/or asymptomatic participants. Overview of results in Table 1 (b). The studies are heterogeneous regarding both purpose and design. IMB is found to play a role in metatarsalgia and especially Morton’s neuroma in several ways: IMB was more frequent in postoperative Morton’s neuroma patients who had persistent pain [10], some studies reported a neuroma-bursa mass [11,12] while others differentiated between IMB and Morton’s neuroma as two independent causes of metatarsalgia [13,14].

#### 3.1.3. Patients with Autoimmune Disorders

Fifteen studies investigated IMB in patients with autoimmune disorders (Table 1 (c)); mostly patients with rheumatoid arthritis, but also patients suspected to develop rheumatoid arthritis, osteoarthritis, or systemic lupus erythematosus. We noted a heterogeneity regarding the bursas investigated, as five of the studies pooled the results of both intermetatarsal bursitis and plantar bursitis.

**Table 1 diagnostics-13-00211-t001:** Study overview.

Author and Year	Type of Study and Design	Aim of Study	Study Population	No. of Patients	Sex (F:M)	Age (Years)
(a) Anatomical studies						
Bossley et al. 1980 [1]	Anatomical	Investigate the role of the intermetatarsal bursa in MN	Cadaveric feet	NA	NA	19–70
Chauveaux et al. 1987 [5]	Anatomical	Describe the anatomic relations of the intermetatarsal bursa	Fresh (<72 h), cadaveric feet	25	NA	NA
Theumann et al. 2001 [15]	Anatomical	Correlate the MRI appearance of the intermetatarsal bursae with the anatomic sections and histopathologic analysis	Fresh, frozen cadaveric feet	8	1:6	78 (75–86)
(b) Patients with metatarsalgia						
Albano et al. 2021 [16]	Diagnostic (US), retrospective	Describe US findings in pts treated with foot orthoses	Pts with metatarsalgia	20	15:5	62.6 (36–78)
Awerbuch et al. 1982 [17]	Treatment, steroid injection and surgery	Treatment of Morton’s metatarsalgia by injected corticosteroid and surgery	Pts diagnosed clinically with MN	50	44:6	48 (28–71)
Bencardino et al. 2000 [18]	Diagnostic (MRI), retrospective	Determine the prevalence of clinically silent MN	Clinically symptomatic (S) and asymptomatic (A) pts with MN on MRI	44 (S: 25, A: 19)	S: 21:4 A: 13:6	S: 49, A: 47
Bossley et al. 1980 [1]	Diagnostic (D) and treatment (T) (anatomic part in section (a))	Investigate the role of the intermetatarsal bursa in MN	D: Pts with forefoot pain and controls and T: Pts with clinical MN	11	NA	NA
Cohen et al. 2016 [11]	Diagnostic (US), prospective	Determine the sonographic appearance of MN	Pts diagnosed with MN having surgery	8	5:5	55 (45–65)
Espinosa et al. 2010 [10]	Diagnostic (MRI), prospective	Evaluate the prevalence of postoperative MRI findings after MN resection	Post-neurectomy pts	58	46:12	45 (27–56)
Hassouna et al. 2007 [14]	Treatment, steroid injection	The clinical effect of US-guided injection in MN	Pts diagnosed clinically with MN	39	32:7	55.8 (26–83)
Iagnocco et al. 2001 [13]	Diagnostic (US), prospective	Identify the changes in the forefoot of pts with metatarsalgia	Pts with monolateral metatarsalgia	112	81:31	58.9 (29–78)
Umans et al. 2014 [19]	Diagnostic (MRI), retrospective	Identify the variety of second and third interspace lesions in relation to plantar plate tears	Pts with metatarsalgia	96	61:35	49
Volpe et al. 1998 [12]	Histological, retrospective	Classify the degree of nerve pathology and compare with the surgical outcome	Pts with MN in the third interspace	32	NA	NA
Zanetti et al. 1997 [20]	Diagnostic (MRI), prospective	Determine the prevalence and size of asymptomatic MN and fluid in the intermetatarsal bursa	Asymptomatic volunteers	70	35:35	45.8 (30–79)
(c) Patients with autoimmune disorders						
Albtoush et al. 2019 [21]	Diagnostic (MRI and US), retrospective	Assess the role of MRI in metatarsalgia and demonstrate the imaging features of IMB by MRI and US	Pts with an autoimmune disorder debuting with metatarsalgia due to IMB	6	6:0	35.8 (20–59)
Bowen et al. 2010 [22]	Diagnostic (US), prospective	Evaluate the effect of anti-TNF therapy on the presence of forefoot pathology, pain, and disability	Pts with RA starting anti-TNF therapy	31	24:7	59.6 (37–76)
Bowen et al. 2010 [23]	Diagnostic (US), prospective	Investigate the prevalence of FFB in US and explore the relationship between FFB and foot impairment	Pts with RA	120	98:22	60.7 ±12.1
Cherry et al. 2014 [24]	Diagnostic (MRI), prospective	To develop a tool for evaluation of FFB at MRI in pts with RA	Pts with RA	30	23:7	61 ±4.1
Dakkak et al. 2020 [25]	Diagnostic (MRI), prospective	To study the prevalence of IMB and SMB in RA pts compared to the other groups	Pts with early (<2 years) RA, other arthritis (O), and healthy controls (C)	RA: 157 O: 284 C: 193	109:48, 158:126, 136:57	59 ± 14, 56 ± 17, 50 ± 16
Dakkak et al. 2020 [26]	Diagnostic (MRI), prospective	Investigate the relationship between physical joint examination and MRI-detected inflammation in MTP joints	Pts with early (<2 years) inflammatory diseases	441	267:174	57 ± 16
Dijk et al. 2021 [27]	Diagnostic (MRI), prospective	Assess the occurrence and prognostic value of IMB	Pts with clinically suspect arthralgia	577	433:144	44 ± 13
Dijk et al. 2021 [28]	Diagnostic (MRI), prospective	Investigate if IMB is a feature of early RA	Pts with early (<2 years) RA	157	109:48	59 ± 14
Dijk et al. 2022 [29]	Diagnostic (MRI), prospective	Investigate if IMB and tenosynovitis contribute to a positive MTP squeeze test	Pts with early (<2 years) RA and pts with clinically suspect arthralgia (S)	RA: 192 and S: 693	102:90, 507:186	57 ± 15 NA
Endo et al. 2018 [30]	Diagnostic (US), case study	Evaluate the use of power Doppler for examining FFB within US	A RA pt with reduced foot mobility	1	1:0	40
Hammer et al. 2019 [31]	Diagnostic (US), retrospective	Investigate the prevalence of IMB and its association with subjective, clinical, and laboratory assessments in RA pts	Pts with RA	209	169:40	53 ± 13
Hooper et al. 2012 [6]	Diagnostic (US), prospective	Describe the natural history of foot- related disability in pts with RA over three years.	Pts with RA	60	51:9	62 (28–89)
Hooper et al. 2014 [32]	Diagnostic (US), prospective	Investigate the prevalence and distribution of FFB in the tree groups	Pts with RA, osteoarthritis, and healthy controls	RA: 56, O: 50, C: 50	NA	41 (20–65) 66.3 (53–80) 62 (28–89)
Koski 1998 [33]	Diagnostic (US), prospective	To detect bursitis in US and correlate findings with symptoms and clinical observations	Pts with RA and healthy controls	RA: 25, C: 30	15:10 18:12	48 (19–74) 45 (13–78)
Mukherjee et al. 2016 [7]	Diagnostic (US), prospective	Determine the prevalence of US-detectable FFB and correlate this to patient-related disability	Pts with systemic lupus erythematosus	20	18:2	53.6 ± 12.8

IMB: intermetatarsal bursitis, MN: Morton’s neuroma, pt(s): patient(s), MRI: magnetic resonance imaging, US: ultrasound, FFB: forefoot bursas, RA: rheumatoid arthritis, C: healthy controls, SMB: Submetatarsal bursitis.

### 3.2. Synthesis of Results

#### 3.2.1. Incidence

The incidence of IMB in patients with metatarsalgia is reported in two studies [13,16]. Iagnocco et al. 2000 report 21.4% (3/14) of examined patients and Albano report 81% (22/27) of examined feet. Hassouna et al. 2007 reported an incidence of IMB in a population with clinically diagnosed Morton’s neuroma and found that 31% (14/45) of patients had IMB [14]. The prevalence of IMB in patients with rheumatoid arthritis varied from 20.6% to 69%, with an average of 40.5% [31,33,34].

#### 3.2.2. Diagnostic Imaging

Of the 28 articles included in this review, 2318 were diagnostic studies. The modalities used were ultrasound and MRI with 11 articles evaluating ultrasound and 10 evaluating MRI. One article from 1980 uses radiography and one uses both MRI and ultrasound [1,21]. The approach to diagnosing IMB varied between studies. All the rheumatic studies with MRI used contrast, while the MRI studies of patients with metatarsalgia were without. For ultrasound, especially the side of the foot scanned varied. Hammer et al. 2019 and Koski 1997 obtained the scans from the dorsal side of the foot while Bowen et al. 2010 scanned from the plantar side [22,31,33]. The rest scanned from both sides. Two of the ultrasound studies used power Doppler [30,31].

Zanetti et al., 1997, investigated asymptomatic intermetatarsal spaces with MRI [20]. They found that 67% of the participants had a fluid collection in at least one intermetatarsal bursa, concluding that fluid collections in the intermetatarsal bursa measuring ≤3 mm should be considered physiologic. Contrary to this, Theumann et al. 2001 note that no bursas were visible on MRI before the intrabursal contrast administration [15]. These contradictory findings might be explained by the fact that one study had living subjects and the other used cadaveric feet.

#### 3.2.3. Clinical Diagnosis

Pain in the intermetatarsal space or metatarsalgia was the phrase used to describe the clinical symptoms of the patients, throughout the included studies. Detailed clinical examination information was limited, and no studies identified symptoms that could be specific to IMB. Albano et al. 2021 provided the most specific description. “To investigate the presence of bursitis, acupressure of metatarsal heads and intermetatarsal spaces was applied and the Mulder test was performed to assess the presence of an MN.” [16] (p. 964). Espinosa et al. 2010 and Hassouna et al. 2007 provided a more detailed description, but it focuses on diagnosing Morton’s neuroma. Dijk et al. 2022 found IMB to contribute to a positive squeeze test in patients with rheumatoid arthritis [29].

#### 3.2.4. Opening Toes

Several articles mention that the spreading of the toes is associated with the presence of IMB in the interspace between the diverting toes. Awerbuch et al. 1982 reported that 45 out of 50 patients had swelling over the painful cleft [17]. Dakkak et al. 2020 reported that 21% of the swollen joints found at physical examination are explained by IMB alone in patients with rheumatoid arthritis [26]. This finding is in line with Koski et al. 1997 who suspected IMB in 5 out of 25 (20%) rheumatoid arthritis patients due to spreading of the toes [33]. Endo et al. 2018 and Hammer et al. 2019 also presented this symptom along with clinical pictures [30,31]. Several terms are used to address this symptom including spreading of the toes, opening toes, and swollen intermetatarsal spaces.

#### 3.2.5. Treatment

Two treatment studies were identified in our search. Awerbuch et al. 1982 treated 50 patients [17]; 28 had complete relief with corticosteroid injections in the intermetatarsal bursa, while 22 had surgery, of which 20 patients had the interdigital nerve and the intermetatarsal bursa removed and two patients only had the interdigital nerve removed. The two patients where the bursa was not removed initially had recurring symptoms and were treated later with either corticosteroid injection or excision of the bursa, both with good results. This is in line with the findings of Espinosa et al. 2010 who showed that IMB is more frequent in symptomatic post-neurectomy patients [10]. Hassouna et al. 2007 treated 39 patients with corticosteroid injections of which 28% had complete relief, but the treatment effect is not evaluated according to ultrasound diagnosis of IMB or Morton’s neuroma, even though it is stated that 31% of the patients had IMB [14].

#### 3.2.6. Autoimmune Disorders

Van Dijk et al. 2022 found IMB to behave and respond to treatment similar to known rheumatoid arthritis characteristics, supporting IMB as an inflammatory feature of rheumatoid arthritis [28]. Bowen et al. 2010 and Cherry et al. 2012 found an association between the presence of forefoot bursas (intermetatarsal and plantar bursas) at ultrasound evaluation and patient-reported disability, at one- and three-year follow-ups of the same cohort [6,23]. IMB is described as associated with the onset of rheumatoid arthritis [17,21,27] (Table 2).

### 3.3. Risk of Bias

Overall, the risk of bias in the included papers was assessed as low but with many unclear risks of bias due to non-reporting of some domains, see Appendix A. A recurring topic of concern was the inconsistency in the definition of forefoot bursas among the articles about autoimmune diseases. Some studies report the presence of both intermetatarsal and submetatarsal bursas under the common name forefoot bursas [7,22,32]. This limits the applicability of the results of these studies.

The predefined criteria for diagnosing IMB varied between the studies. There were differences in the location of the bursa in relation to the deep transverse metatarsal ligament [7] and four studies did not state which criteria they used to diagnose IMB [14,18,21,33]. The lack of predefined criteria causes a high risk of bias score under the index test domain. Regarding reference standards, we marked studies with unclear risk of bias, if they did not use a reference standard. Studies that used a pathological evaluation of surgical or dissection specimens received a low risk of bias evaluation, similar to the approach in systematic reviews about Morton’s neuroma [35,36].

## 4. Discussion

This study shows that IMB must be considered a cause of pain in patients with metatarsalgia, as well as patients with rheumatic diseases. In patients with a complaint of forefoot pain and the presence of spreading toes, IMB should be a diagnostic consideration. The identified studies address various aspects of the intermetatarsal bursa and IMB. There is agreement on the presence and location of the intermetatarsal bursas and the close relationship between the neurovascular bundle and the distal part of the intermetatarsal bursas.

The reported incidence of IMB range from 20.6% to 93.3% [23,31]. The incidences must be compared cautiously as very few of the studies are comparable due to a great variance in the study population and design. Some studies report the incidence based on the number of patients and others on the number of feet and some report incidences that include both intermetatarsal and submetatarsal bursitis. Additionally, different methods are applied in diagnostic studies to obtain ultrasound and MRI scans with the purpose of diagnosing IMB. Several of the included studies mention the clinical sign opening toes/swollen intermetatarsal spaces/V-sign in relation to IMB [17,30,31]. Widening of the intermetatarsal angle on radiographs, which we interpret as representing the same symptom, have also been linked to the presence of Morton’s neuroma, with a sensitivity between 30–73% [37,38]. No studies have investigated both conditions and the presence of opening toes, but the widening of the intermetatarsal space should raise clinical suspicion of both conditions.

The criteria for diagnosing IMB with ultrasound is described as “IMB is detected as a hypoechoic structure between the metatarsal heads” by Hammer et al. 2019 and by Albano et al. 2021 as “a hypo to anechoic fluid collection in the intermetatarsal space” [16,31]. Compared with the ultrasonic criteria for Morton’s neuroma: “The typical sonographic appearance is that of an ovoid, hypoechoic mass oriented parallel to the long axis of the metatarsals”, Redd et al. 1989 and by Oliver et Beggs 1998 “a round or oval hypoechoic mass was found between the metatarsal heads” [39,40]. The diagnostic criteria for IMB and Morton’s neuroma by ultrasound are very similar with the presence of a hypoechoic mass between the metatarsal heads. This might result in an overestimation of the condition one is looking for due to the similar ultrasonic presentation.

Diagnostic criteria for IMB on MRI include a contrast-enhanced lesion above the deep transverse ligament and between the metatarsal heads, either with or without rim enhancement [26]. Due to an inconsistent contrast enhancement pattern of Morton’s neuroma on MRI, Zanetti et al. 2005 recommended not to use contrast-enhanced images for MRI of Morton’s neuroma [41]. This causes a limitation in the exclusion of IMB or differentiation between an intermetatarsal bursa with fluid and IMB. The diagnostic criteria for Morton’s neuroma on MRI are a well-demarcated lesion at the level of the neurovascular bundle on the plantar side of the deep transverse metatarsal ligament, with a signal intensity that is low to intermediate on T1-weigthed images and hypointense to fatty tissue on T2-weigthed images [42,43]. Some studies report the presence of Morton’s neuroma, which occurs hyperintense on T2-weigthed images or extends dorsal and past the deep transverse metatarsal ligament [44,45]. This inconsistency in imaging results is noticeable—could these hyperintense/dorsal extending neuromas represent cases of intermetatarsal bursitis?

Cohen et al. 2016 suggested a common name to describe the conditions: neuroma-bursal complex, as their results suggested the two conditions to be coherent [11]. They observed a discrepancy between ultrasound and specimens’ measures similar to the results of Read et al. 1999 [11,46]. At histologic examination, both studies found that the nerve thickening was surrounded by scarred bursal tissue and they both described an intimate relationship between bursa and neuroma at theultrasound. We find this observation correlates with the Grade 1 Morton’s neuroma described by Volpe et al. 1998 [12]. Quinn et al. 2000 also questions the sonographic diagnosis of Morton’s neuroma due to the location of the intermetatarsal mass and they mentioned the intermetatarsal bursa as a cause [47]. A review article by Bianchi et al. 2014 described the target for steroid treatment for Morton’s neuroma to be the bursa, but also addresses IMB as an independent disease [48]. Overall this raises the questions: Is IMB a contributing factor in the development of Morton’s neuroma, and which condition is associated with the symptoms experienced by the patients? Diagnostic accuracy studies about Morton’s neuroma often use histological evaluation of surgical specimens as a reference standard. Some studies have questioned the value of the histological findings in specimens from Morton’s neuroma surgery due to the near 100% hit rate [49,50]. Pathologic studies of interdigital nerves from asymptomatic feet find macro- and microscopic alterations compatible with those of Morton’s neuroma—more commonly appearing with proceeding age, thereby concluding that Morton’s neuroma does not have a recognizable morphological substrate [51,52]. Four imaging studies have investigated the incidence of asymptomatic Morton’s neuromas [18,20,53,54]. With incidences of 25–53% (average 35.5%) alterations that fulfil the diagnostic criteria for Morton’s neuromas in asymptomatic patients are common and in line with the findings in the pathologic studies. This leads us to raise the question: Is IMB the cause of pain in the intermetatarsal spaces and Morton’s neuroma a painless, degenerative development of the interdigital nerve?

The mechanisms behind the development of IMB are not directly discussed in the included papers, except for Dijk et al. 2021 who found IMB to be linked with early rheumatoid arthritis. Looking at other types of bursitis (olecrani, iliopsoas, ect.), they are caused by mechanical forces such as trauma or overuse. In line with this, mechanical strain as a cause of IMB provides a reasonable explanation [55,56]. Rheumatoid arthritis is a risk factor for the development of bursitis—especially for early arthritis, as it is for IMB [57]. Our study has limitations that should be addressed. Descriptions of IMB is often a short paragraph in papers with another main focus, and some of these papers do not mention intermetatarsal bursa/bursitis in the title or abstract. This results in a selection bias, as these studies have not been included in our study.

The risk of bias in included studies adds to the risk of bias in our results. We found an inconsistency in the definition of forefoot bursas and in the stated predefined criteria to diagnose IMB. These inconsistencies might reflect the limited written knowledge of IMB and because of this, each paper reflects the authors’ own experience instead of a more uniform approach. Another limitation is that many of the included studies are more than 20 yearsold (10 out of 28). Due to technological advancement in the intermediate period, the conclusions from these studies might have been different in a current setting [36,58]. Among the included diagnostic studies, very few had a reference standard. The frequent use of conservative treatment for patients with metatarsalgia and medicine for patients with autoimmune disorders all limit the possibility for specimens for histopathological evaluation [16,22].

The lack of a reference standard combined with the notable difference in the diagnostic criteria for IMB, and advancement in the diagnostic technology might partly explain the dissemination of the results presented in this review article. This emphasizes a need to develop proper diagnostic criteria for IMB. IMB is yet to be definitively categorized as a differential diagnosis or a contributing factor in the development of Morton’s neuroma.

## 5. Conclusions

IMB must be considered a cause of pain in patients with metatarsalgia and patients with rheumatic diseases. In patients with forefoot pain and the presence of spreading toes/V-sign, IMB should be a diagnostic consideration.

Future diagnostic imaging studies would benefit from protocols that can differentiate IMB from other causes of forefoot pain—especially Morton’s neuroma, with the use of contrast in MRI studies and power Doppler in ultrasound studies. Furthermore, care should be taken in the differentiation between the physiologic fluid in the intermetatarsal bursa and IMB. Increased focus on IMB could aid a better understanding of its role in forefoot pain and has the potential to improve treatment strategies in the future.

## Figures and Tables

**Figure 1 diagnostics-13-00211-f001:**
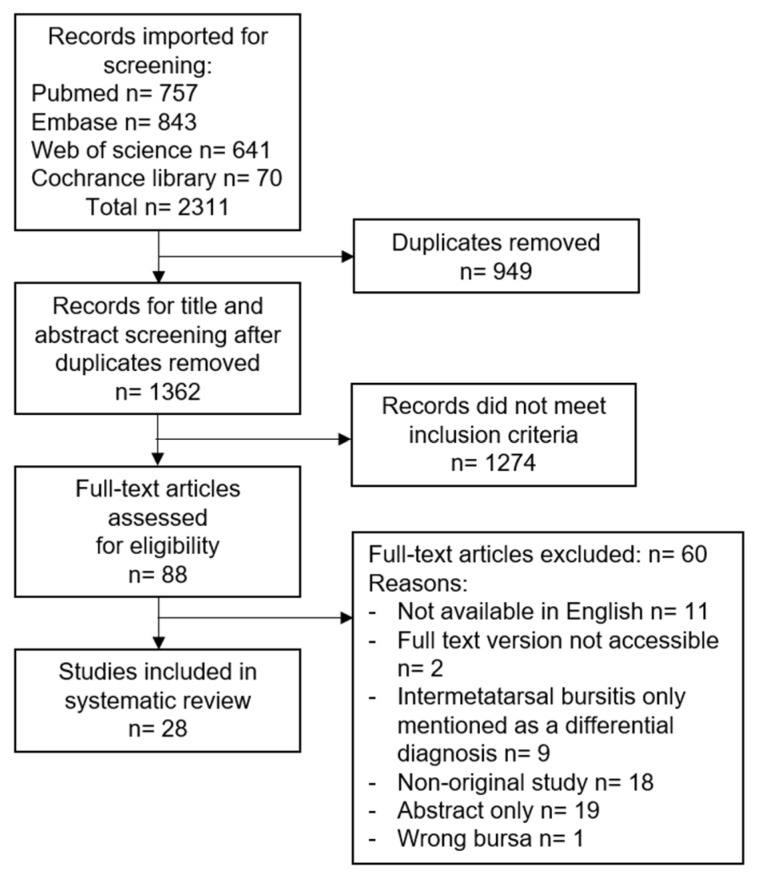
Review process overview.

**Table 2 diagnostics-13-00211-t002:** Summary of results.

Incidence of IMB	21.4–81% of patients with metatarsalgia 31% of patients clinically diagnosed with MN 20.6–69% of patients with RA, average 40.5%
Diagnostic imaging	23 diagnostic studies 11 ultrasound 10 MRI - Studies of rheumatic patients used contrast, *n* = 6 Studies of patients with metatarsalgia did not use contrast, *n* = 4 1 both MRI and ultrasound 1 radiography
Clinical diagnosis	Not systematically investigated
Opening toes	Presence of swelling or spreading toes was reported in 20-90% of cases
Treatment	IMB is associated with persistent symptoms after surgical treatment of MN
Autoimmune disorders	IMB is frequent in the early stages of RA, responds in line with other RA symptoms to treatment and is associated with patient-reported disability

## Data Availability

Not applicable.

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
