# Peer review of "Diagnostic Considerations of Intermetatarsal Bursitis: A Systematic Review"

_diagnostics, 2023, doi:10.3390/diagnostics13020211_

Round 1

Reviewer 1 Report

This article fulfilled some vacant area in diagnostic research of intermetatarsal bursitis. Yet the mechanism hasn't been interpreted very clearly. There have been heated discussions on diagnostic criteria of IMB, but so far manifestations of this particular disease have seldom been linked to its immune mechanism. Authors could add more paragraphs in the Discussion part about this concern.

Author Response

Point:

Yet the mechanism hasn't been interpreted very clearly. There have been heated discussions on diagnostic criteria of IMB, but so far manifestations of this particular disease have seldom been linked to its immune mechanism. Authors could add more paragraphs in the Discussion part about this concern.

Response:

Thank you for your feedback and interest.

We have added a paragraph to the discussion section regarding the mechanisms behind intermetatarsal bursitis and how it behaves similar to other types of bursitis. Hope this addresses your comment.

Paragraph added (page 11, line 281-286):

“The mechanisms behind the development of IMB are not directly discussed in the included papers, except for Dijk et al. 2021 who found IMB to be linked with early rheumatoid arthritis. Looking at other types of bursitis (olecrani, iliopsoas, ect.), they are caused by mechanical forces such as trauma or overuse. In line with this, mechanical strain as a cause of IMB provide a reasonable explanation. Rheumatoid arthritis is a risk factor for the development of bursitis – especially for early arthritis, as it is for IMB”.

Reviewer 2 Report

Ottimo articolo.

Numero sufficiente di casi

Essendo un Ch. Toracico volevo chiedere ai Autori se per caso abbiano osservato su Rx dei pz. nodi al livello del parenchima polmonare?

Author Response

Point:

Essendo un Ch. Toracico volevo chiedere ai Autori se per caso abbiano osservato su Rx dei pz. nodi al livello del parenchima polmonare?

Response:

Thank you for your feedback.

As our Italian language skills are very limited, our answer is based on a translation of your comment. To our knowledge the problematics around diagnosing intermetatarsal bursitis, and especially differentiating it from Morton’s neuroma, is very specific for the feet. Therefore, we have not looked into diagnostics of the lungs.

We wonder if the comment migth be for a different article?

Reviewer 3 Report

This is a systematic review of a topic of relative interest to medical specialists in this area. The methodology is correct and the results are adequately presented.

My only recommendation is:

- Tables 1 and 2 are of no interest to the reader and, for my part, it would be more recommendable if they became complementary tables.

- On the contrary, the information of interest corresponds to section 3.3 "Summary of results". I think it is convenient that a summary table be shown with what is mentioned in that section, which is what is really useful for the reader.

Author Response

Point 1:

Tables 1 and 2 are of no interest to the reader and, for my part, it would be more recommendable if they became complementary tables.

Response 1:

Thank you for your feedback and suggestions.

Regarding table 1 - we agree that it is a bit inaccessible due to its size. However, in our opinion, the overview of current literature serves a main role in the paper to see an outline of previous studies concerning design, patientgroup ect., for the sake of future studies.

Regarding table 2 – we agree, and the table has been moved to supplementary material.

Point 2:

On the contrary, the information of interest corresponds to section 3.3 "Summary of results". I think it is convenient that a summary table be shown with what is mentioned in that section, which is what is really useful for the reader.

Response 2:

Thank you for the idea to present the results in a more convenient way. We have made a table summarizing the results from section 3.2 Synthesis of results (added to the manuscript on page 8 just following section 3.2):

Table 2. Summary of results
Incidence of IMB

21.4-81% of patients with metatarsalgia

31% of patients clinically diagnosed with MN

20.6-69% of patients with RA, average 40.5%

Diagnostic imaging

23 diagnostic studies

11 ultrasound

10 MRI         

  • Studies of rheumatic patients used contrast, n=6      
  • Studies of patients with metatarsalgia did not use contrast, n=4

1 both MRI and ultrasound1 radiography

Clinical diagnosis Not systematically investigated

Opening toes

Presence of swelling or spreading toes was reported in 20-90% of cases
Treatment IMB is associated with persistent symptoms after surgical treatment of MN
Autoimmune disorders IMB is frequent in the early stages of RA, responds in line with other RA symptoms to treatment and is associated with patient-reported disability